# Structure Design, Kinematics Analysis, and Effect Evaluation of a Novel Ankle Rehabilitation Robot

**Shuwei Qu [1], Ruiqin Li [1],\*, Wei Yao [2],\*, Chunsheng Ma [1] and Zhihong Guo [1]**

[1] School of Mechanical Engineering, North University of China, Taiyuan 030051, China; shuweiqu1222@nuc.edu.cn (S.Q.); machunsheng@nuc.edu.cn (C.M.)

[2] Department of Biomedical Engineering, University of Salford, Glasgow G4 0NW, UK

\* Correspondence: liruiqin@nuc.edu.cn (R.L.); w.yao@salford.ac.uk (W.Y.)

**Abstract:** This paper presents a novel ankle rehabilitation (2-CRS+PU)&R hybrid mechanism, which can meet the size requirements of different adult lower limbs based on the three-movement model of the ankle. This model is related to three types of movement modes of the ankle movement, without axis offset, which can cover the ankle joint movements. The inverse and forward position/kinematics results analysis of the mechanism is established based on the closed-loop vector method and using the optimization of particle groups algorithm. Four groups of position solutions of the mechanism are obtained. The kinematics simulation is analyzed using ADAMS software. The variations of the velocity and acceleration of all limbs are stable, without any sudden changes, which can effectively ensure the safety and comfort of the ankle model end-user. The dexterity of the mechanism is analyzed based on the transport function, and the results indicate that the mechanism has an excellent transfer performance in yielding the structure parameters. Finally, the rehabilitation evaluation is conducted according to the three types of movement modes of the ankle joint. The results show that this ankle rehabilitation mechanism can provide a superior rehabilitation function.

**Keywords:** ankle rehabilitation; (2-CRS+PU)&R hybrid mechanism; kinematics; dexterity; rehabilitation effect evaluation

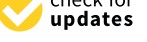



## 1. Introduction

The ankle joint is a key driving force in the human body. Therefore, the ankle joint is the most vulnerable to damage, which can seriously affect people's walking ability and daily life.

Several experimental studies on market products of ankle rehabilitation robots have been reported in the literature. Du et al. [1] proposed an ankle rehabilitation robot based on the 3-RRS spherical parallel mechanism. However, the axis of the kinematic pairs must have a high machining and assembly accuracy. This makes the project costs particularly high, which makes it difficult to promote in the market. Chang et al. [2] proposed an RRR-PaRPS-RHJ parallel mechanism for ankle rehabilitation with three rotation degrees of freedom to overcome the rehabilitation motion decoupling. Obviously, the stiffness and carrying capacity of the decoupled configuration are far less than those of the parallel/hybrid mechanism. Tsoi et al. [3] replaced the middle passive link with the lower limb of the patient in a parallel ankle robot. This design can match the anatomical ankle joint by placing four actuators above the end-effector (AaEE), but unexpected loads may be exerted, causing discomfort and safety issues. Yu et al. [4] proposed one single DOF 3R open-chain, non-circular gear-linkage mechanism. It is only suitable for ankle plantar flexion and back extension, and it cannot realize the movement requirements of adduction/abduction and valgus/valgus of the ankle joint. Liao et al. [5] proposed a hybrid ankle rehabilitation robot. It greatly improves the movement space and dexterity of the mechanism. Wang et al. [6] designed two special serial spherical mechanisms as the constraint limb to construct several 3-DOF parallel mechanisms for addressing the issue of non-coincidence of the centers

through redundantly actuated guarantees of no singularity, better dexterity, and enhanced stiffness within the prescribed workspace. Zou et al. [7] proposed a 3-RRS lower-mobility, parallel ankle rehabilitation mechanism. This mechanism can rotate around the rotation center to match the ankle motion mechanism. Saglia et al. [8] proposed a redundantly actuated parallel mechanism, wherein the actuated method eliminated the configuration singularities and increased the dexterity. Dai et al. [9] developed several parallel mechanisms with a central strut for ankle rehabilitation. The central strut configuration enhanced the mechanism's stiffness. Jamwal et al. [10] proposed a soft parallel robot for ankle joint rehabilitation through a genetic algorithm, with an optimal robot global conditioning number. Deng et al. [11] adopted a neural network algorithm for control of hydraulic manipulators.

Girone et al. [12] realized a 6-DOF Stewart platform-based system for ankle telerehabilitation control, able to be monitored remotely. However, the system is very complex for the mechanism with 6-DOF. Zhang et al. [13] analyzed a reconfigurable workspace and the torque capacity in order to improve the ankle rehabilitation robot's comfort and safety issues. Mustafa et al. [14] proposed a fuzzy logic control scheme for a 2-DOF redundantly actuated parallel ankle rehabilitation robot. Tursynbek et al. [15] presented a framework for the analysis of a 3-RRR spherical parallel manipulator with coaxial input axes (coaxial SPM), with a focus on its infinite rotational movement capabilities and its effects on the manipulator's characteristics.

It can be noted that the existing ankle rehabilitation robots have some limitations. First, most robots only have three degrees of freedom (DOFs) according to the ankle motion model, and they cannot be used for the size requirements of different adults' lower limbs. Second, for most spherical parallel robots, their rotation centers can easily shift away from the rotation center of the human ankle [1,6,16]. Such a shift can cause an uncomfortable feeling in the patients and can further cause a secondary injury of the ankle joint. Third, evaluations of patients' comfort, safety, and training effects are lacking.

An ankle rehabilitation robot with all degrees of ankle movements and without axis offset is needed to greatly improve the safety in rehabilitation movements. In this work, we propose a novel hybrid manipulator for ankle rehabilitation based on a 2-CRS+PU parallel mechanism, which can generate rotation analogous to human ankles without axis offset and provide all degrees of freedom according to the ankle movement mode. Meanwhile, the robot can meet the size requirements of different adults' lower limbs.

The remainder of this paper is organized as follows. Section 2 proposes the type synthesis and structure design based on the ankle joint movement model. Section 3 analyzes the kinematics, including inverted solutions and the Jacobian matrix. Section 4 simulates the performance of the mechanism used for the rehabilitation exercise. Section 5 analyzes the dexterity, with the relevant parameters. Section 6 evaluates the patients' comfort and rehabilitation effects using AnyBody software (Thaitu Software Technology Co., LTD, 2019 (Shangai, China)). Finally, conclusions are drawn in Section 7.

## 2. Type Synthesis and Structure Design

### 2.1. The Movement Modes of the Human Ankle

Ankle joint movements include plantar flexion and dorsiflexion movements around the frontal axis ($x$ axis), inversion and eversion movements around the sagittal axis ($y$ axis), and the abduction and adduction movements around the vertical axis ($z$ axis). A coordinate system was established at the center of the ankle socket, taking a human's left foot as an example, as shown in Figure 1.

The movement range of the ankle joint is limited by ligaments to protect the ankle from excessive damage, which are shown as Table 1 [17].

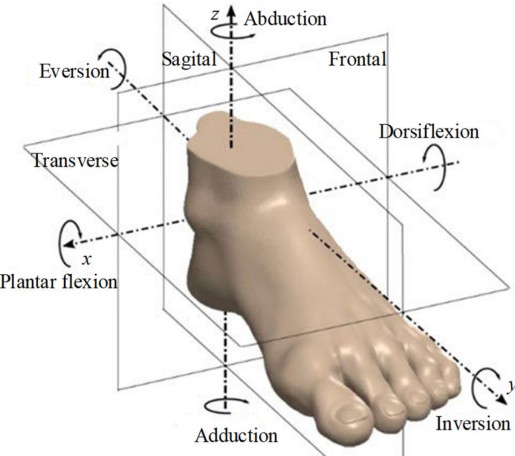

**Figure 1.** The movements of the ankle joint.

**Table 1.** The movement ranges of human ankle joint.

| Rotation Direction | Movement Model | Movement Range |
|---|---|---|
| Around $X$-axis | Plantar flexion | 40–45° |
| | Dorsiflexion | 25–30° |
| Around $Y$-axis | Abduction | 15° |
| | Adduction | 35–40° |
| Around $Z$-axis | Eversion | 25–30° |
| | Inversion | 25–30° |

*2.2. Type Synthesis of the Mechanism*

Ankle rehabilitation mechanism should be universal, and meet the size requirements of different adults' lower limbs and meanwhile satisfy three-dimensional rotation. Therefore, the mechanism can be considered as three rotations around the corresponding axes and one translation degree of freedom along the $z$ axis. Its movement mode is the rotation about three mutually perpendicular intersecting axes. Therefore, the ankle rehabilitation mechanism should be capable of three-dimensional rotation. In addition, the rotation center should be coincident with the ankle joint center so as to avoid second injury in rehabilitation process.

Supposing the movement mode is $R^xR^yR^zT^z$, according to the movement and constraint relationship of the screw theory [18], the expected number of degrees of freedom is $M(w,p)$, where $w$ and $p$ denote the number and the property of the degree of freedom required. Let the screw system of the mechanism be $\left\{ \$_M \middle| (\$_1^m, \$_2^m, \cdots, \$_w^m) \right.$. This means that if $\$_M$ is known, $M(w,p)$ can be determined. By solving the reciprocal screw of $\$_M$, the reciprocal screws of the screw system of the mechanism can be determined. The reciprocal screws of the screw system of the limbs $\left\{ \$_r^{lj} \middle| (\$_{j1}^{lr}, \$_{j2}^{lr}, \cdots, \$_{ji}^{lr}) \right.$ (where $i \leq 6-w$ is the number of the reciprocal screws of the *j*-th *j*th limb) are a subset of those of the mechanism, and the relation can be described as follows.

$$
\begin{aligned}
M(w,p) &\Leftrightarrow \left\{ \$_M \middle| \$_1^m, \$_2^m, \cdots, \$_w^m \right\} \Leftrightarrow \\
&\left\{ \$_r \middle| S_1^r, \$_2^r, \cdots, \$_{6-w}^r \right\} \Leftrightarrow \\
&\left\{ \$_{lj}^r \middle| \$_{j1}^r, \$_{j2}^r, \cdots, \$_{ji}^r \right\} \Leftrightarrow \\
&\left\{ \$_{lj}^m \middle| \$_{j1}^m, \$_{j2}^m, \cdots, \$_{j(6-i)}^m \right\} \Leftrightarrow \{PMs\}
\end{aligned}
\tag{1}
$$

which must satisfy

$$
\begin{cases}
\left\{ \boldsymbol{S}_{j1}^r, \boldsymbol{S}_{j2}^r, \cdots, \boldsymbol{S}_{ji}^r \right\} \subseteq \left\{ \boldsymbol{S}_1^r, \boldsymbol{S}_2^r, \cdots, \boldsymbol{S}_{6-w}^r \right\} \\
\overset{k}{\underset{j=1}{\cup}} \left\{ \boldsymbol{S}_{j1}^r, \boldsymbol{S}_{j2}^r, \cdots, \boldsymbol{S}_{ji}^r \right\} = \left\{ \boldsymbol{S}_1^r, \boldsymbol{S}_2^r, \cdots, \boldsymbol{S}_{6-w}^r \right\} \\
\overset{k}{\underset{j=1}{\cap}} \left\{ \boldsymbol{S}_{lj}^m \middle| \boldsymbol{S}_{j1}^m, \boldsymbol{S}_{j2}^m, \cdots, \boldsymbol{S}_{j(6-i)}^m \right\} = \left\{ \boldsymbol{S}_1^m, \boldsymbol{S}_2^m, \cdots, \boldsymbol{S}_w^m \right\}
\end{cases}
\tag{2}
$$

In order to ensure no axis deviation during ankle rehabilitation, synthesize one constrained limb PU (P is prismatic pair, U is universal joint), and make the rehabilitation mechanism realize two rotations and one movement; then, synthesize two active unconstrained CRS limbs (C, R, and S are cylinder pair, revolute pair, and spherical pair, respectively). 2-CRS+PU hybrid mechanism can rotate around the platform $x$ axis, $y$ axis, and translate along the fixed platform $z$ axis; synthesize one rotation around the platform along $z$ axis by a revolute pair R. Therefore, the mechanism can realize three rotations around three orthogonal axes and one translation along the $z$ axis.

### 2.3. Structure Design

During the ankle rehabilitation, the axis is always at one point and gives a passive constraint chain PU. The mechanism can realize the two rotations and one translation movement. The other two active unconstrained chains are both of six degrees of freedom. In order to save the cost and facilitate installation of motors, the CRS type of the active unconstrained chains are selected. Therefore, the ankle rehabilitation mechanism consists of a base, a hybrid platform, a 2-CRS+PU parallel mechanism, and an adjustable foot pedal. The (2-CRS+PU)&R hybrid ankle rehabilitation mechanism is shown in Figure 2.

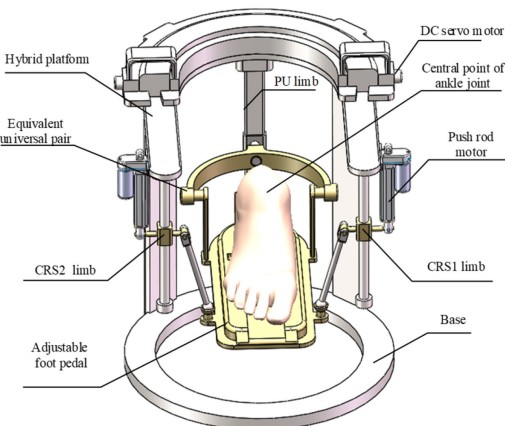

**Figure 2.** Ankle rehabilitation robot.

The degree of freedom of the 2-CRS+PU parallel mechanism is the rotation around the platform $x$, $y$ axes and translation along the $z$ axis. The (2-CRS+PU)&R hybrid rehabilitation mechanism can realize three revolution and one translation (3R1T) movements by mixing a revolute pair (R) around the platform $z$ axis. In order to convenience all limbs and keep the rehabilitation mechanism coinciding with the ankle center in the rehabilitation process, the universal joint U is designed as two orthogonal revolute pairs and through the central point of the ankle joint as shown in Figure 2.

## 3. Kinematic Analysis

### 3.1. Inverse Position Analysis

An equivalent 2-CRS+PU parallel mechanism is established, as shown in Figure 3. The moving platform $\Delta B_1 B_2 B_3$ and the fixed platform $\Delta A_1 A_2 A_3$ are all equilateral triangles, and their circumcircle radii are $2a$ and $2b$, respectively.

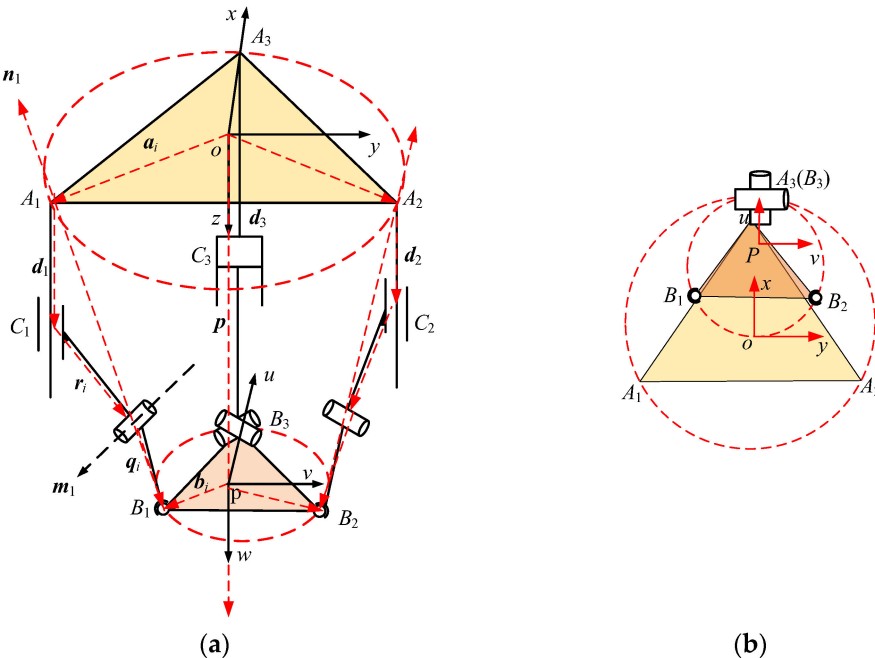

**Figure 3.** 2-CRS+PU parallel mechanism. (**a**) Coordinate system. (**b**) Geometric relationship of the moving and fixed platform.

The fixed coordinate system {*O-xyz*} is fixedly connected to the fixed platform, and its coordinate origin is located at the geometric center *O* of the fixed platform. The moving coordinate system {*P-uvw*} is fixedly connected to the moving platform, and its coordinate origin is located at the geometric center *P* of the moving platform. *y* axis and *v* axis are parallel to $A_1A_2$ and $B_1B_2$, respectively; the *x* axis and *u* axis coincide with $OA_3$ and $PB_3$, respectively; the direction of *w* axis and *z* axis is according to right hand rule, as shown in Figure 3.

The orientation of the moving platform {*P-uvw*} relative to fixed coordinate system {*O-xyz*} can be expressed as rotation matrix *R*. We define three rotation angles *α*, *β*, *γ* as roll, pitch, and yaw from the moving platform to the fixed platform. Since the 2-CRS-PU parallel mechanism does not rotate around the *w* axis, *γ* = 0°. The rotation matrix R of the moving coordinate system {*P-uvw*} relative to the fixed platform coordinate system {*O-xyz*} can be simplified as

$$^{O}\mathbf{R}_p = \mathbf{Rot}_{(u)}(\alpha)\mathbf{Rot}_{(v)}(\beta) \tag{3}$$

The rotation matrix contains sine and cosine functions of the variable rotation angles *α*, *β*, and *γ*. Referring to Figure 3, a vector closure equation for each limb *i* can be written as

$$\boldsymbol{a_i} + \boldsymbol{d_i} + \boldsymbol{r_i} + \boldsymbol{q_i} = \boldsymbol{p} + \boldsymbol{b_i}\ (i = 1, 2) \tag{4}$$

where $\boldsymbol{a_i}$, $\boldsymbol{p}$ and $\boldsymbol{b_i}$ are the position vectors of the fixed and moving platform expressed in the fixed reference frame; $\boldsymbol{q_i}$, $\boldsymbol{r_i}$ and $\boldsymbol{q_i}$ are the *i*-th limb vectors.

The third limb satisfied the constraint

$$\boldsymbol{a_3} + \boldsymbol{d_3} = \boldsymbol{p} + \boldsymbol{b_3} \tag{5}$$

In the plane of the *i*-th (*i* = 1, 2) limb, vector $\boldsymbol{m_i}$ is always perpendicular to vector $\boldsymbol{n_i}$. Therefore,

$$\boldsymbol{m_i^T} \cdot \boldsymbol{n_i} = 0 \tag{6}$$

where $\boldsymbol{m_i}$ is axial vector of the *i*-th rotation pair; $\boldsymbol{n_i}$ is the direction vector of $B_i$ to $A_i$.

The position parameters of the fixed platform point $A_i$ in the coordinate system $\{O\text{-}xyz\}$ can be expressed as

$$^oA_1 = (-\sqrt{3}a, -a, 0), \ ^oA_2 = (\sqrt{3}a, -a, 0), \ ^oA_3 = (0, 2a, 0)$$

The position parameters of the moving platform point $B_i$ in the coordinate system $\{P\text{-}uvw\}$ can be expressed as

$$^pB_1 = (-\sqrt{3}b, -b, 0), \ ^pB_2 = (\sqrt{3}b, -b, 0), \ ^pB_3 = (0, 2b, 0)$$

According to the coordinate transformation formula, there is

$$^oB_i = {}^oR_P{}^PB_i + {}^oP_P \tag{7}$$

where $^oB_i$ is point $B_i(i = 1, 2, 3)$ position in the coordinate system $\{O\text{-}xyz\}$, and $^oP_P$ is the position vector from $P$ to $O$.

According to the closed-vector relation, there is

$$\begin{cases} a_i + d_i + q_i + r_i + s_i = p + b_i \\ a_3 + q_3 = p + b_3 \end{cases} \tag{8}$$

From the constraints relationship,

$$m_3^T \cdot q_3 = 0 \tag{9}$$

The geometric position relationship of the moving platform and fixed platform moving satisfied the constraint relationship as follows:

$$\begin{cases} m_i = (^oB_i - {}^oB_{i+1}) \times A_i \\ n_i = {}^oB_i - {}^oA_i \\ m_3 = \begin{bmatrix} 1 & 0 & 0 \end{bmatrix}^T \\ q_3 = {}^oB_3 - {}^oA_3 \end{cases} \quad (i = 1, 2) \tag{10}$$

Bring Equation (10) into Equations (8) and (9), and then there is

$$\begin{cases} d_1 = \left(\sqrt{3}a + 2b\sin\beta\sin\alpha - \sqrt{3}b\cos\beta + b\sin\beta\sin\alpha, \ a + y + b\cos\alpha, \ z + \sqrt{3}b\sin\beta + b\cos\beta\sin\alpha\right)^T \\ d_2 = \left(-\sqrt{3}a + 2b\sin\beta\sin\alpha + \sqrt{3}b\cos\beta + b\sin\beta\sin\alpha, \ a + y + b\cos\alpha, \ z - \sqrt{3}b\sin\beta + b\cos\beta\sin\alpha\right)^T \\ d_3 = (-2a + y - 2b\cos\alpha, \ z - 2b\cos\beta\sin\alpha)^T \end{cases} \tag{11}$$

where

$$y = \frac{(-\sqrt{3}b\cos\beta + 3b\sin\beta\sin\alpha)(\sqrt{3}a - \sqrt{3}b\cos\beta + b\sin\beta\sin\alpha)}{3b\cos\alpha} - \frac{2b\sin\beta\sin\alpha + 3b\cos\alpha + \sqrt{3}b\sin\beta - 3b\cos\beta\sin\alpha}{3b\cos\alpha}$$

$$\begin{cases} |d_1| = \sqrt{\left(\sqrt{3}a - \sqrt{3}b\cos\beta + 3b\sin\beta\sin\alpha\right)^2 + h^2 + \left(\sqrt{3}b\sin\beta + b\cos\beta\sin\alpha + z\right)^2} \\ |d_2| = \sqrt{\left(-\sqrt{3}a + \sqrt{3}b\cos\beta + 3b\sin\beta\sin\alpha\right)^2 + h^2 + \left(-\sqrt{3}b\sin\beta + b\cos\beta\sin\alpha + z\right)^2} \\ |d_3| = \sqrt{(3 - 3b\cos\alpha + h)^2 + (z - 2b\sin\alpha\cos\beta)^2} \end{cases} \tag{12}$$

where

$$h = \frac{(\sqrt{3}b\cos\beta + 3b\sin\beta\sin\alpha)(\sqrt{3}a - \sqrt{3}b\cos\beta + b\sin\beta\sin\alpha) - 2b\sin\beta\sin\alpha}{3b\cos\alpha} - \frac{(-\sqrt{3}b\sin\beta + 3b\cos\beta\sin\alpha)(z + \sqrt{3}b\sin\beta + b\cos\beta\sin\alpha)}{3b\cos\alpha}$$

Let $2a = 300$ mm, $2b = 150$ mm. Given 4 groups, pose parameters of the moving platform $\alpha$, $\beta$. The driving parameters $d_1$, $d_2$, $d_3$ (pretended by $L_1$, $L_2$, $L_3$) can be solved according to Equation (12).

The inverse solutions are obtained, as shown in Table 2.

**Table 2.** Numerical examples of the inverse solutions.

| Pose | $\alpha/(°)$ | $\beta/(°)$ | $z$/mm | $L_1$/mm | $L_2$/mm | $L_3$/mm |
|------|------|------|------|------|------|------|
| 1 | 0 | 0 | 180 | 48.5 | 48.5 | 208.5 |
| 2 | 0 | 10 | 200 | −43.6 | −56.6 | 207.4 |
| 3 | 15 | 0 | 200 | 19.7 | 23.6 | 202.5 |
| 4 | 15 | 10 | 263 | 37.5 | 59.6 | 222.9 |

*3.2. Forward Position Analysis*

Given input parameters ($L_1$, $L_2$, $L_3$), solve the output orientation ($\alpha$, $\beta$, $z$). The solution algorithm procedure is shown in Figure 4.

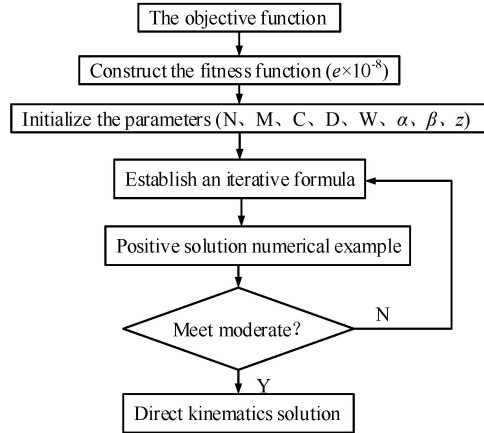

**Figure 4.** Procedure of the forward position algorithm.

The objective function is Equation (12); the fitness function of the particle group is constructed; and the minimum iteration value shall converge within $e \times 10^{-8}$. The fitness function is constructed as follows.

$$F_{min} = min\sum_{i=1}^{3}|F_i| = |F_1| + |F_2| + |F_3| < e \times 10^{-8} \tag{13}$$

Initialize the parameters, N = 40, M = 800, C = 2, D = 3, W = 0.8, $\alpha \in (-45°, 45°)$, $z \in (0°, 150\,\text{mm})$.

Establishing the iteration update formula and velocity iteration update formula, the position $X_t$ and velocity $V_t$ of the particle at any moment can be expressed as follows.

$$\begin{cases} X_t = X_{t-1} + V_t \\ V_t = W \cdot V_{t-1} + C_1 \times rand \times (O_t - X_t) + C_2 \times rand \times (P_t - X_t) \end{cases} \tag{14}$$

Given the displacement parameters $L_1$, $L_2$, $L_3$ of 4 sets of driving pairs, the final forward position solution is shown in Table 3.

**Table 3.** Numerical examples of positive solutions.

| Pose | $L_1$/mm | $L_2$/mm | $L_3$/mm | $\alpha/°$ | $\beta/°$ | $z$/mm |
|------|------|------|------|------|------|------|
| 1 | 48.5 | 48.5 | 202.5 | 0.050 | 0.076 | 180.5 |
| 2 | −47.6 | 57.8 | 197.3 | 0.240 | 4.817 | 197.6 |
| 3 | 19.7 | 29.7 | 204.5 | 4.844 | 0.363 | 191.5 |
| 4 | 30.5 | 59.5 | 201.6 | 4.948 | 5.124 | 262.9 |

Table 3 shows that the positive solution calculation after entering the displacement parameter approaches the exact value.

The deviations of positive solution between numerical and algorithm are shown in Table 4.

**Table 4.** Deviations of positive solution numerical algorithm.

| Pose | $|\Delta\alpha|/°$ | | $|\Delta\beta|/°$ | $|\Delta z|/$mm |
|:---:|:---:|:---:|:---:|:---:|
| 1 | 0.050 | 0.076 | 0.076 | 0.352 |
| 2 | 0.240 | 0.183 | 0.183 | 0.250 |
| 3 | 0.157 | 0.363 | 0.363 | 0.152 |
| 4 | 0.052 | 0.124 | 0.124 | 0.166 |

Table 4 shows that the maximum deviation of the displacement parameter of the positive solution value calculation case is 0.363°, which verifies the correctness of the positive solution.

The fitness change curves of the above four sets of positive solution examples are shown in Figure 5.

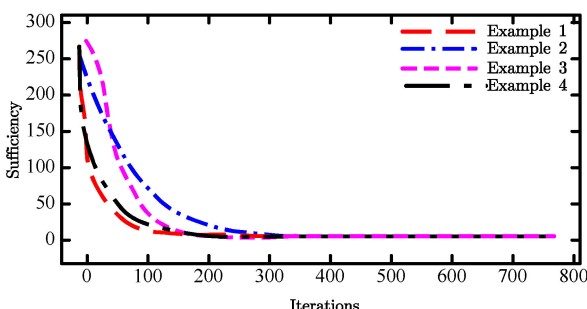

**Figure 5.** Positive solution algorithm of fitness curve.

Figure 5 shows that the fitness change curves of the four sets of positive solution examples converge fast, and the four groups of curves almost converge to 0 about 350 times, which indicates that the particle swarm algorithm adopted has high accuracy and meets the accuracy requirements.

## 4. Kinematics Simulation and Analysis

The 3D model is imported into Adams, as shown in Figure 6. The kinematics simulation is performed according to the movement characteristics of the ankle joint.

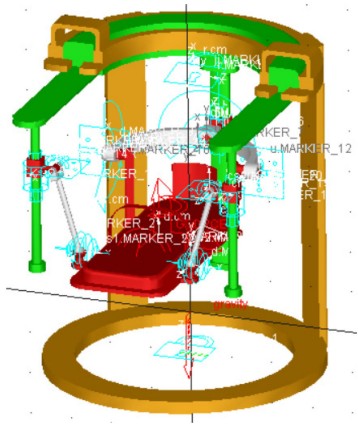

**Figure 6.** Virtual prototype model in ADAMS software environment.

### 4.1. Plantar Flexion and Dorsiflexion Movements

According to the step drive function of plantar flexion and dorsiflexion rehabilitation movement range, the displacement change of the moving platform angle and the driving

limb of the mechanism is shown in Figure 7. The change trend of the moving platform angular velocity and the velocity of the driving limb are shown in Figure 8.

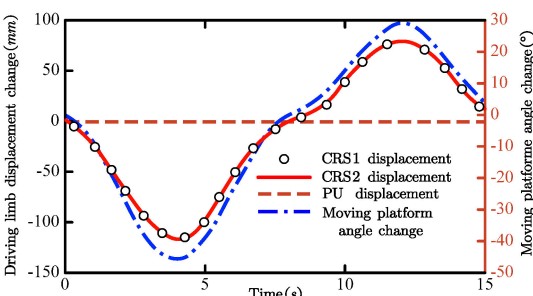

**Figure 7.** Characteristics of moving platform and driving limb.

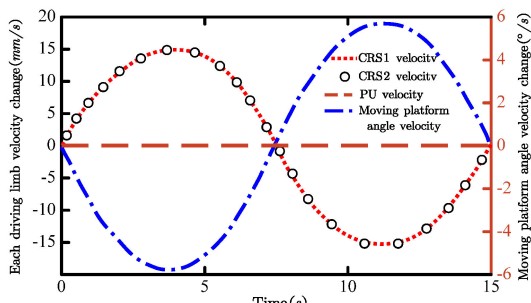

**Figure 8.** Variations of angular velocity and limb linear velocity.

Figure 7 shows that the displacement of the ankle arch is $-45°$, and the displacement of CRS1 and CRS2 limbs is $L_1 = L_2 = -121.9$.

When the ankle reaches $30°$, the displacement of CRS1 and CRS2 is $L_1 = L_2 = 83.27$. Therefore, when the movement ranges of plantar flexion and dorsiflexion of the ankle joint is $[45°, 30°]$, the range of displacement change corresponding to each driving limb of the mechanism is

$$-121.9 \leq L_1 = L_2 \leq 83.27, L_3 = 0 \tag{15}$$

Figure 8 shows that for the plantar flexion and dorsiflexion movements, the maximum velocity of the driving limb is 15 mm/s, and the maximum angular velocity of the moving platform is $6°/s$. The velocity is gently and has no impact, which can ensure the safety of the rehabilitation movement.

### 4.2. Inversion and Eversion Movements

According to the step driver function of the inversion and eversion recovery movement, the kinematic simulation is conducted. The change trend of the platform angle and the displacement of each driving limb is shown in Figure 9. The variation trend of the platform angular velocity and the velocity of each driving limb are shown in Figure 10.

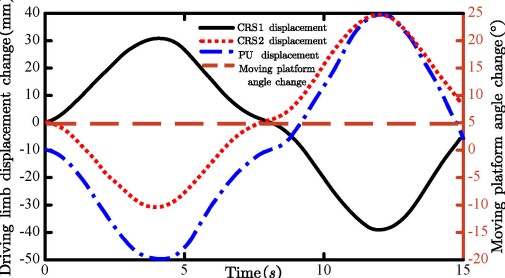

**Figure 9.** Characteristics of the moving platform and the driving limbs.

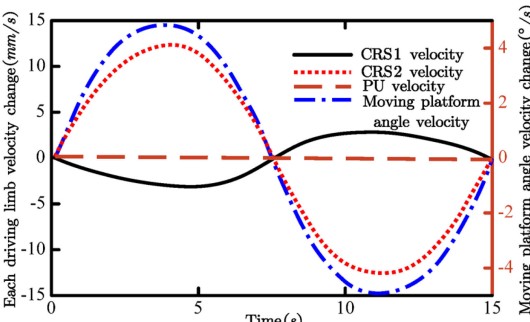

**Figure 10.** Variation of the moving platform and the driving limbs.

Figure 9 shows that when the ankle inversion is at $-20°$, the CRS1 limb is $L_1 = 31.14$, and CRS2 limb is $L_2 = -31.78$, and PU limb is $L_3 = 0$; when at $25°$, the CRS1 limb is $L_1 = -40.53$, CRS2 limb is $L_2 = 39.16$, and PU limb is $L_3 = 0$. When the movement range of the ankle inversion and eversion joint is $[-20°, 25°]$, the displacement change range corresponding to each driving limb of the mechanism is

$$\begin{cases} -40.53 \le L_1 \le 31.14 \\ -31.78 \le L_2 \le 39.16 \\ L_3 = 0 \end{cases} \tag{16}$$

Figure 10 shows that when the mechanism does inversion and eversion movement, the maximum velocity of the driving limb is 12 mm/s, and the maximum angular velocity of the moving platform is 15°/s. The velocity is gentle and without any impact, which can ensure the safety of the rehabilitation movement.

### 4.3. Adduction and Abduction Movements

According to the step driving function, the variation trends of the angle and angular velocities of the moving platform are shown in Figure 11.

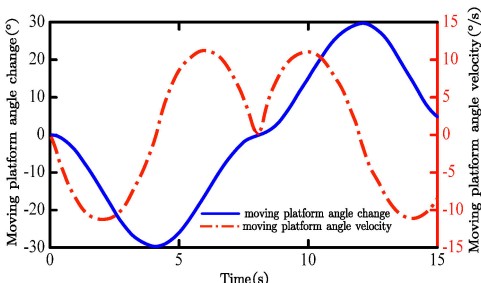

**Figure 11.** Variations of the angles and angular velocities of the moving platform.

Figure 11 shows that the mechanism realizes adduction and abduction movements of the ankle joint through the hybrid rotation of the revolute pair R. The maximum movement range is $[-30°, 30°]$, and the change range of angular velocity of the moving platform is $[-11.25°/s, 11.25°/s]$, satisfying the range of movement and safety requirements during ankle rehabilitation.

The simulation results of Figures 7 and 9 are identical with the inverse kinematic solution of Table 2 and forward kinematic solution of Table 3.

## 5. Dexterity Analysis

Dexterity represents the mapping relationship of the mechanism input/output and is an important indicator of mechanism transfer performance. The better the dexterity of the mechanism is, the higher the accuracy of the mechanism is, the more reasonable the structure design is, and the better the transmission performance is.

### 5.1. Dexterity of Plantar Flexion and Dorsiflexion

The radii of the fixed platform and the moving platform of 2-CRS+PU parallel mechanism are

$$2a = 300 \text{ mm}, 2b = 150 \text{ mm} \tag{17}$$

The transmission function of the plantar flexion and dorsiflexion mechanism is characterized by the rotational transmission performance along the $u$ axis. The condition number evaluation index and the operability evaluation index are shown in Figures 12 and 13, respectively.

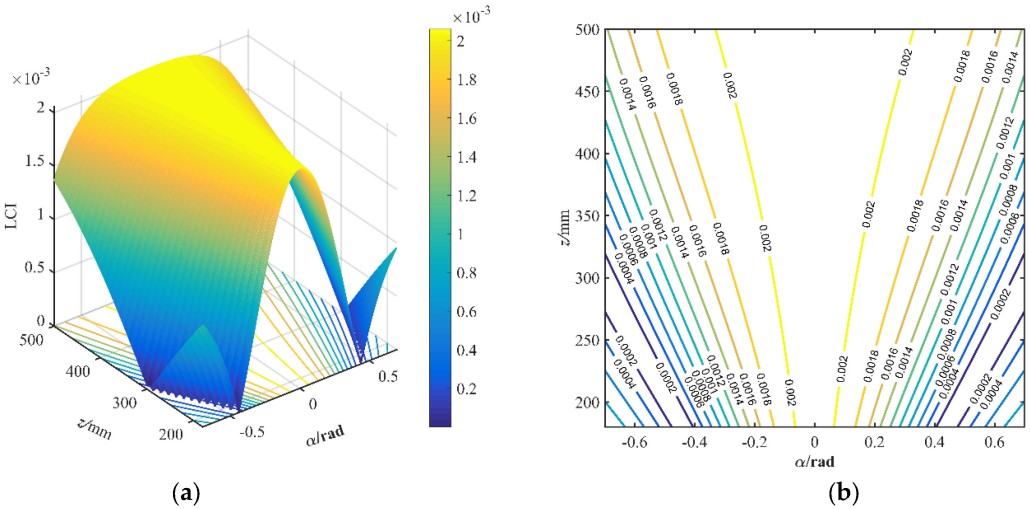

|       |       |
| :---: | :---: |
| (**a**) | (**b**) |

**Figure 12.** Local condition number of plantar flexion and dorsiflexion. (**a**) Local condition number surface. (**b**) Local condition number contour.

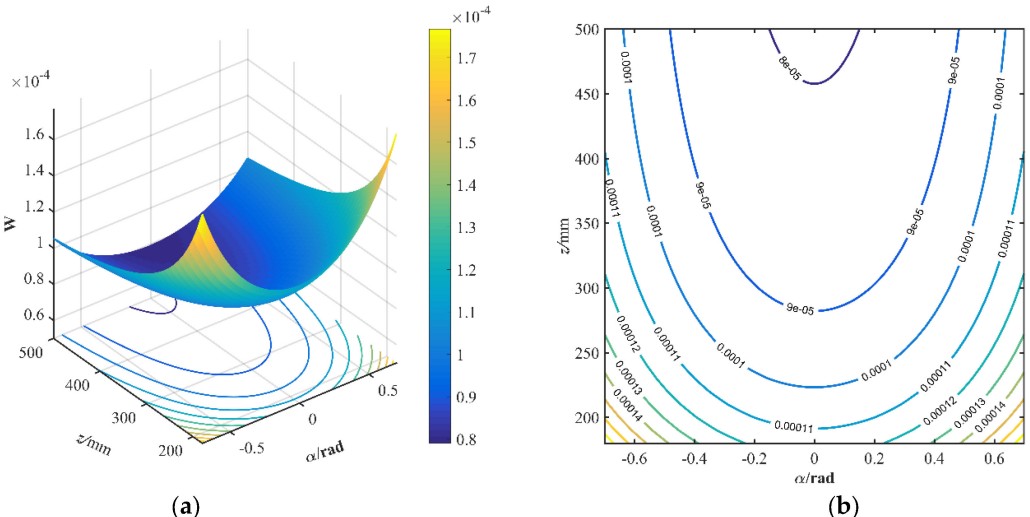

|       |       |
| :---: | :---: |
| (**a**) | (**b**) |

**Figure 13.** Operability number of plantar flexion and dorsiflexion. (**a**) Operability number surfaces. (**b**) Operability number contour.

Figure 12 is the dexterity index of local condition number about $\alpha$ and $z$ when $\beta = 0$. The figure is symmetrical at $\alpha = 0$ rad, and the mechanism local condition number has the most value 0.002 at $\alpha = 0$ rad; the larger the value of $z$ is, the slower the trend of the local condition number is. Therefore, the $z$ value should be relatively large when the plantar flexion and dorsiflexion rehabilitation movements around the $u$ axis.

Figure 13 is the operability number of plantar flexion and dorsiflexion about $\alpha$ and $z$ when $\beta = 0$. Figure shows that the farther from $\alpha = 0$ rad, the smaller the value of $z$ is, and the greater and slowing the trend of the conditions number is; when $z$ is [200 mm, 250 mm]

and $\beta$ is [−0.8 rad, −0.4 rad] or [0.4 rad, 0.8 rad], the value of operability $W$ is large, and the dexterity performance is better.

Therefore, when the ankle rehabilitation mechanism ensures the optimal transmission performance of plantar flexion and dorsiflexion rehabilitation exercises around the $u$ axis, the value of $z$ of 2-CRS-PU parallel mechanism should be relatively large. The values of $z$ and $\beta$ should satisfy the following range.

$$200 \text{ mm} \leq z \leq 250 \text{ mm},$$
$$-0.8 \text{ rad} \leq \beta \leq -0.4 \text{ rad} \text{ or } 0.4 \text{ rad} \leq \beta \leq 0.8 \text{ rad} \tag{18}$$

### 5.2. Dexterity of Adduction and Abduction

The transmission function characteristic of the adduction and abduction are around the $v$ axis. The condition number evaluation index and the operability evaluation index are shown in Figures 14 and 15.

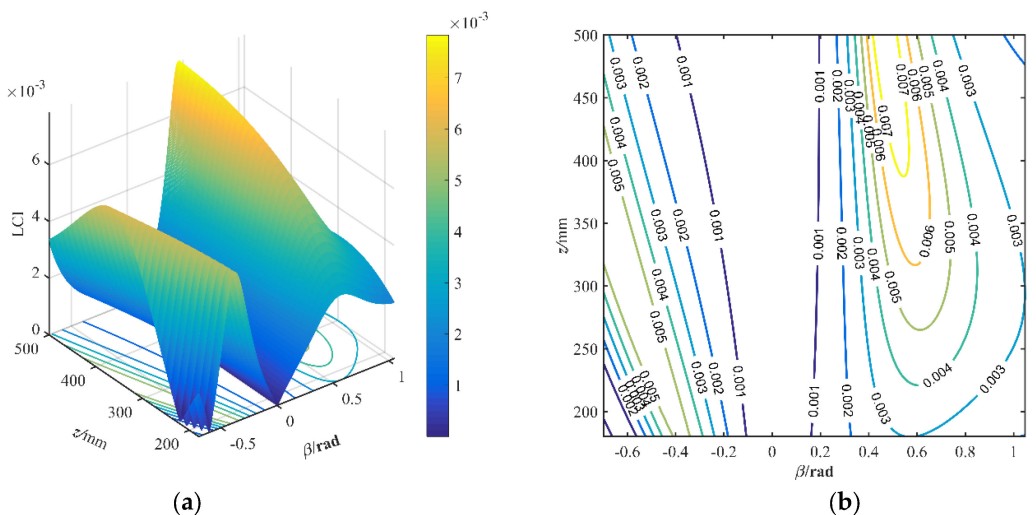

**Figure 14.** Condition number of adduction and abduction. (**a**) Local condition number surfaces. (**b**) Local condition number contour.

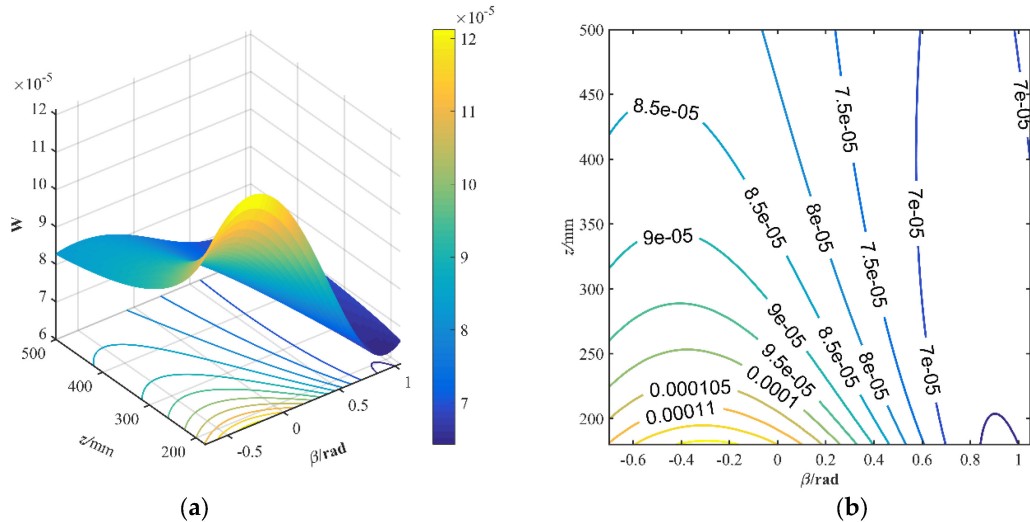

**Figure 15.** Operability number of abduction and adduction. (**a**) Operability number surfaces. (**b**) Operability number contour.

Figure 14 is the dexterity index of local condition number about $\beta$ and $z$ when $\alpha = 0$. Figure 14 shows that the local condition number is the largest when $\beta = 0.5$ rad; the value

range of $\beta$ is $[-1$ rad, 0 rad$]$, and the mechanism dexterity performance is the maximum at $\beta = -0.5$ rad; Figure 15 shows the operability $W$ is the maximum when $\beta = -0.2$ rad and the optimal value range of $\beta$ is $[-0.6$ rad, 0 rad$]$; when $z$ is the range $[200, 250]$, the value of the operability $W$ is generally large, and the dexterity is relatively excellent.

Therefore, when the ankle rehabilitation mechanism ensures the best transmission performance, the $z$ value is within $[300, 50]$, and the $\beta$ value is within $[-0.6$ rad, 0 rad$]$.

## 6. Rehabilitation Exercises

Based on the man–machine coupling model, the rehabilitation movements are simulated using AnyBody software. The rehabilitation effects are evaluated by the amount of activity and changes of muscle strength. The kinematics simulation was loaded, and the ankle rehabilitation mechanism (2-CRS+PU)&R completed one cycle rehabilitation exercise. The simulation results under different rehabilitation exercises extreme position are shown in Figure 16.

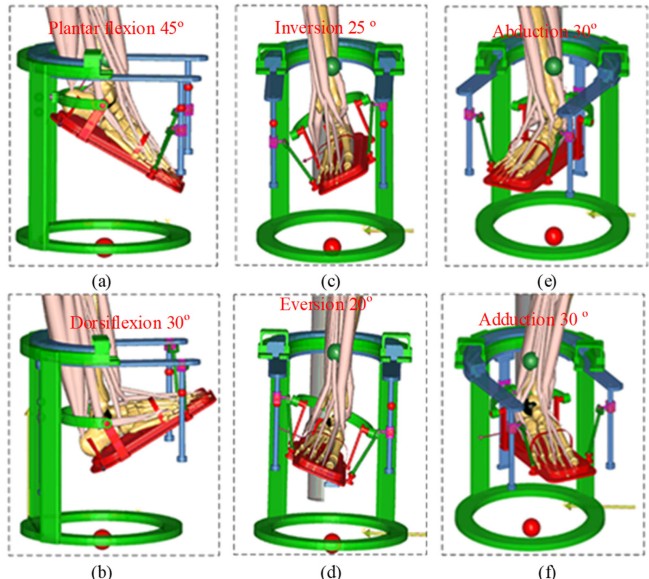

**Figure 16.** Simulated extreme position of ankle rehabilitation exercise.

### 6.1. Plantar Flexion and Dorsiflexion Rehabilitation Exercise

Figure 16 shows that the plantar flexion rehabilitation exercise can reach $45°$; the dorsiflexion rehabilitation exercise can reach $30°$.

The amount of activity and changes of the muscle strength in plantar flexion and dorsiflexion rehabilitation exercise are shown in Figures 17 and 18 respectively.

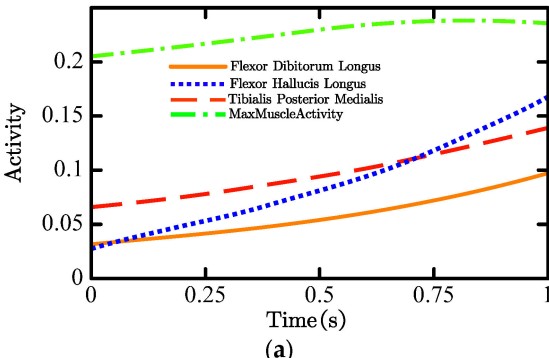

**Figure 17.** *Cont.*

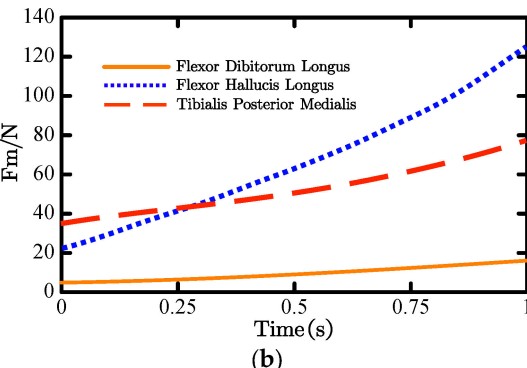

**Figure 17.** Plantar flexion and dorsiflexion exercise. (**a**) The amount of activity of plantar flexion rehabilitation exercise. (**b**) The muscle strength of plantar flexion rehabilitation exercise.

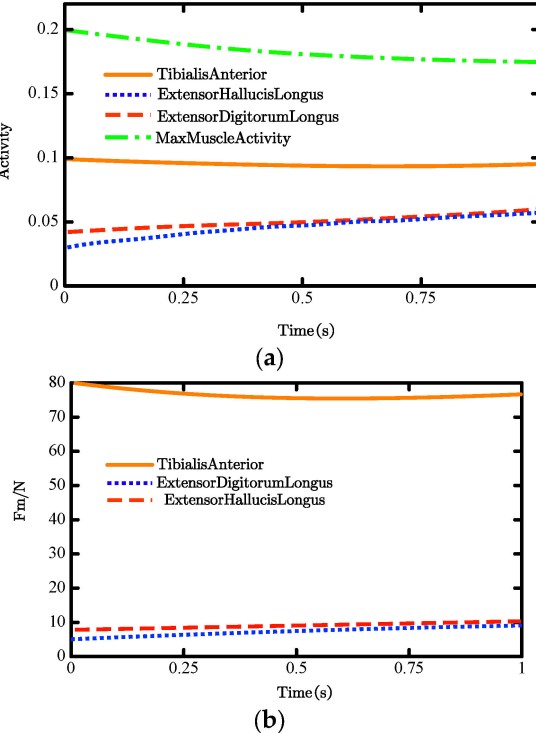

**Figure 18.** Dorsiflexion rehabilitation exercise. (**a**) The amount of activity of dorsiflexion rehabilitation exercise. (**b**) The muscle strength of dorsiflexion rehabilitation exercise.

Figure 17a shows that the maximum activity of the flexor longus and flexor hallucis was 0.09663 and 0.1655, while the maximum value was less than 0.25, much less than the maximum activity 1 during muscle strain. The results indicate that the ankle muscle activity was effectively activated, and there is no secondary strain. Figure 17b shows the muscle force changes of flexor longus, flexor hallucis, and tibialis posterior within 1 s are 10.541 N, 103.14 N, and 40.52 N, respectively, in which the muscle force of flexor hallucis changes is the most, indicating that the ankle muscle strength is greatly strengthened at this time, and the ankle rehabilitation effect is obvious.

Figure 18a shows that the maximum activity of the toe extensor, thumb extensor, and tibial anterior muscles were 0.06009, 0.06009, and 0.1054 under dorsiflexion rehabilitation exercises, in which the maximum activity of the tibialis anterior muscles and the maximum value were all less than 0.21, far less than the limit activity of 1. The result indicates that the ankle muscle activity received effective recovery and no secondary strain. Figure 18b shows the muscle force changes of the toe extensor, toe extensor, and anterior tibialis within 1 s are 3.802 N, 2.533 N, and −1.92 N, respectively, among which the muscle force changes

of the toe extensor are the most, indicating that the strength of the ankle muscle is slightly strengthened at this time, and the ankle rehabilitation effect is good.

### 6.2. Inversion and Eversion Rehabilitation Exercise

Figure 16 shows that the inversion rehabilitation exercise can reach 30°; the eversion rehabilitation exercise can reach 30°.

The amount of activity and changes in muscle strength of inversion and eversion exercises are shown in Figures 19 and 20, respectively.

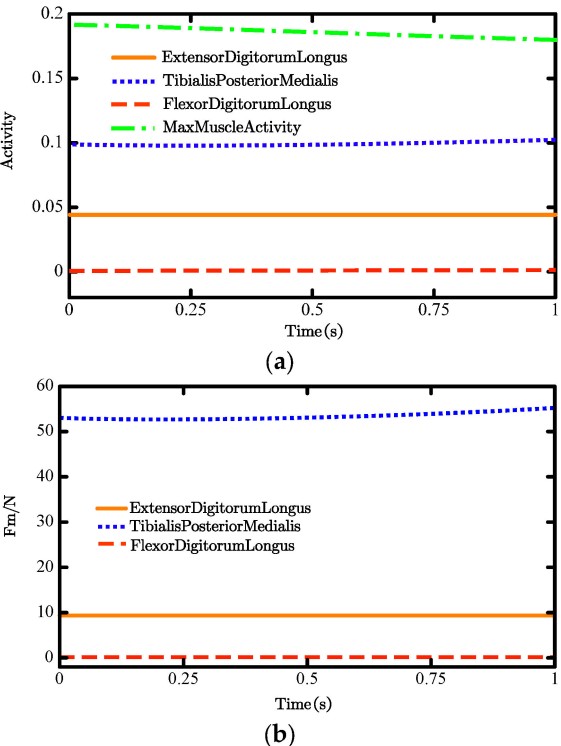

**Figure 19.** Inversion rehabilitation exercise. (**a**) Inversion rehabilitation exercise amount of activity. (**b**) The muscle strength of inversion rehabilitation exercise.

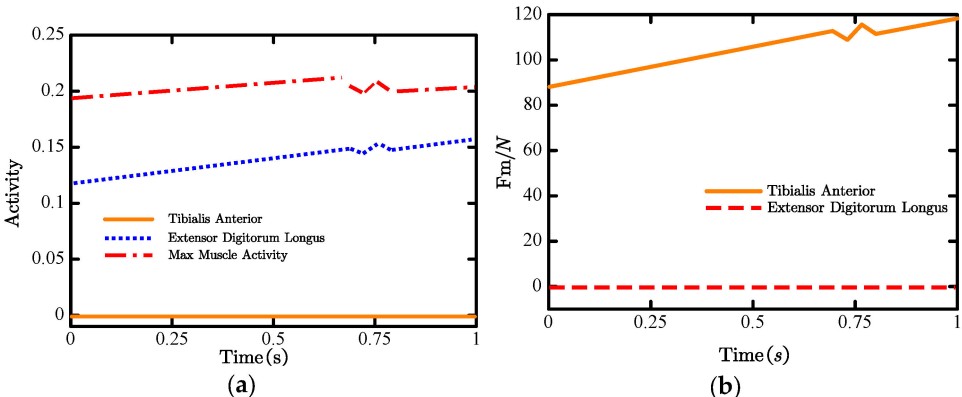

**Figure 20.** Eversion rehabilitation exercise. (**a**) The amount of activity of eversion rehabilitation exercise. (**b**) The muscle strength of eversion rehabilitation exercise.

Figure 19a shows that the maximum activity of the long flexor, long extensor, and posterior tibial muscles were approaching 0, 0.05763, and 0.1023, and the maximum activity of the posterior tibial muscles and the maximum value were all less than 0.20, far less than the limit activity 1. The result indicates that the ankle muscle activity was effectively activated, and the ankle recovered without secondary strain. Figure 19b shows that the

muscle force changes of the long flexor, long extensor, and posterior tibialis within 1 s are approaching 0 N, 0.105 N, and 2.46 N, respectively, among which the muscle force changes of the posterior tibialis are the most, indicating that the ankle muscle strength is slightly strengthened at this time, and the ankle rehabilitation effect is good.

Figure 20a shows that under the eversion rehabilitation exercise, the maximum values of the tibial anterior tibia and toe extensor activity were close to 0 and 0.1623, and the maximum value of the ankle muscle activity was 0.25, far less than the limit amount of muscle strain 1, which indicates that the ankle has effective recovery. Figure 20b shows that the muscle force changes of the tibialis anterior muscle and the toe long extensor within 1 s are close to 0 N and 28.45 N, respectively, among which the tibialis anterior muscle changes the most, indicating that the strength of the ankle muscle is greatly strengthened at this time, and the rehabilitation effect of the ankle joint is obvious.

### 6.3. Adduction and Abduction Rehabilitation Exercise

Figure 16 shows that the adduction rehabilitation exercise can reach 25°; the abduction rehabilitation exercise can reach 20°.

The amount of activity and changes in muscle strength of the adduction and abduction exercises are shown in Figures 20 and 21, respectively.

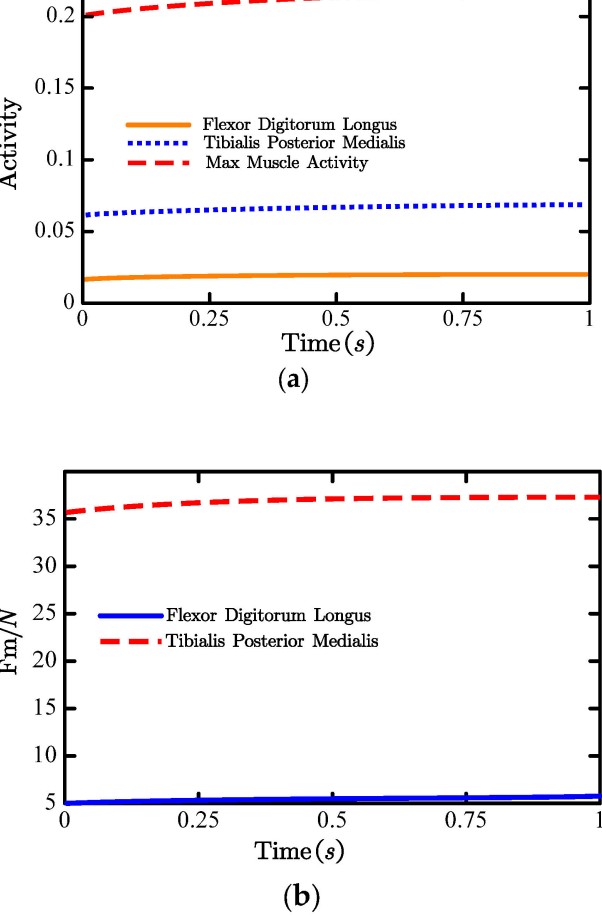

**Figure 20.** Adduction rehabilitation exercise. (**a**) The amount of activity of adduction rehabilitation exercise. (**b**) The muscle strength of adduction rehabilitation exercise.

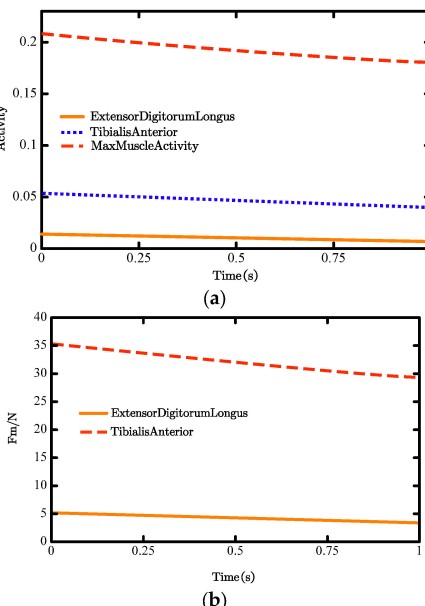

**Figure 21.** Abduction rehabilitation exercise. (**a**) The amount of activity of the abduction rehabilitation exercise. (**b**) The muscle strength of the abduction rehabilitation exercise.

Figure 20a shows that the maximum activity of the posterior tibial muscle and toe long flexors was 0.06976 and 0.03366. The maximum value of the ankle muscle activity was less than 0.22, far less than the limit activity of 1, which indicates that the ankle joint achieved effective recovery without secondary strain. Figure 20b shows that the muscle force changes of the posterior tibialis muscles and long flexor muscles within 1 s were 2.54 N and 0.365 N, respectively, among which the muscle force changes of the posterior tibialis muscles changed the most. The result indicates that the strength of the ankle muscles was slightly strengthened at this time, and the ankle rehabilitation effect was good.

Figure 21a shows that the maximum activity of the tibial anterior muscle and toe long extensor was 1.0.0651 and 0.03142, in which the tibialis anterior activity was the largest, while the ankle muscle activity was less than 0.21, far less than the limit activity of muscle strain 1. The result indicates that the ankle had effective recovery and no secondary strain. Figure 21b shows that the muscle stress changes of the tibialis anterior muscle and the toe long extensor within 1 s are −6.17 N and −1.386 N, respectively, among which the tibialis anterior muscle stress changes the most, indicating that the strength of the ankle muscles is slightly strengthened at the time, and the ankle rehabilitation effect is good.

The cycle rehabilitation exercise results showed that the ankle muscle activity was activated; the muscle contraction performance was improved; the muscle strength was strengthened; and the damaged ankle joint was effectively recovered.

## 7. Conclusions

The (2-CRS+PU)&R hybrid ankle rehabilitation mechanism can avoid axis offset by splitting the universal joint U as two axes orthogonal revolute pairs and through the central point of the ankle joint.

The inverse and forward movement solutions are analyzed. The curves approximately converged to 0 at iterations around 350 times, which indicates that the adopted optimization of the particle groups algorithm method has high accuracy and meets the accuracy requirements.

Kinematic simulations are performed according to the characteristics of plantar flexion and dorsiflexion, inversion and eversion, and adduction and abduction movements of the ankle joint in Adams software environments. The maximum velocity of the driving limb is 15 mm/s; the maximum angular velocity of the moving platform is 15°/s. The velocity is gentle and without any impact, which can ensure the safety of the rehabilitation movement.

Dexterity is analyzed according to the movement model through local condition number and operability number. Both indicators are basically the same, which further indicates the validity of the dexterity indicators. The kinematic simulations and dexterity analysis results show that the (2-CRS+PU)&R hybrid mechanism has an excellent transmission performance.

The man–machine coupling model was constructed using AnyBody software. The mechanism can meet the range of movement according to the pattern of ankle movement. Moreover, the activity amount and muscle strength under different rehabilitation exercises were activated, which shows that the ankle muscles were effectively exercised during rehabilitation, and the ankle rehabilitation performed well.

**Author Contributions:** Conceptualization, S.Q.; methodology, S.Q.; software, C.M.; validation, S.Q.; formal analysis, R.L.; investigation, W.Y.; resources, S.Q.; data curation, Z.G.; writing—original draft preparation, S.Q.; writing—review and editing, R.L.; visualization, W.Y.; supervision, R.L.; project administration, Z.G.; funding acquisition, S.Q. All authors have read and agreed to the published version of the manuscript.

**Funding:** This work were supported by Fundamental Research Program of Shanxi Province (Nos. 20210302124220,202203021211101), Key R&D projects in Shanxi Province (Nos. 201803D421028, 201903D421051), and Shanxi Scholarship Council of China (No. 2021-114).

**Institutional Review Board Statement:** Not applicable.

**Informed Consent Statement:** Not applicable.

**Data Availability Statement:** Not applicable.

**Conflicts of Interest:** The authors declare no conflict of interest.

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
