# Peer review of "Structure Design, Kinematics Analysis, and Effect Evaluation of a Novel Ankle Rehabilitation Robot"

_applsci, doi:10.3390/app13106109_

Round 1

Reviewer 1 Report

Please provide a comprehensive analysis of existing ankle rehabilitation robot configurations, detail their respective characteristics, and explain the advantages of the design you have created.

Author Response

Reviewer 1

Please provide a comprehensive analysis of existing ankle rehabilitation robot configurations, detail their respective characteristics, and explain the advantages of the design you have created.

Answer:

    Authors carefully studied all the cited references, from the configuration structure, motion performance, the feasibility of marketing comprehensive analysis of existing ankle rehabilitation robot. Authors proposed the innovation of the research work of this manuscript from the existing ankle rehabilitation robot limitations.

Reviewer 2 Report

My comments are:

1. The literature review should be much enriched to highlight the novelty of this paper. 

2. The main contribution of this paper is not clear enough. It should be summarized to improve the readability. 

3. Some related research might be mentioned and discussed in the introduction section, such as 'neural network-based adaptive asymptotic prescribed performance tracking control of hydraulic manipulators'. 

Author Response

Reviewer 2

1.The literature review should be much enriched to highlight the novelty of this paper.

Answer:

Authors have analyzed and commented on the ankle mechanisms citied in the introduction, which were marked in introduction.

  1. The main contribution of this paper is not clear enough. It should be summarized to improve the readability.

Answer:

Author summarizes and improves the main innovation work and results of this, and specifically it from the abstract and conclusions.

  1. Some related research might be mentioned and discussed in the introduction section, such as 'neural network-based adaptive asymptotic prescribed performance tracking control of hydraulic manipulators'

Answer:

Control algorithm was mentioned and this article has been citied.

Reviewer 3 Report

This paper presents a novel ankle rehabilitation 2-CRS+PU hybrid mechanism, related to 8 three types of movement modes of the ankle movement without axis offset which can cover the ankle joint movement. However, there are some content that could be improved.

1.This paper proposes a foot and ankle rehabilitation robot, presents some work, but the innovations in the paper are not clear. It is suggested to present it in the introduction with subheadings.

2. Please explain the title of 3.1 (and 3.2)? Is it an inverse kinematic analysis?

3. It is suggested that the black curve in Figure 18 in the paper be in another color or line type, which is easily misunderstood. (Other figures as well)

4. The description in line 404 of the text guarantees safety. It seems that the current results or conclusions in the paper is hard to support this conclusion.

5. In line 410, the author says "The results show that the ankle muscles were effectively exercised during rehabilitation and the ankle rehabilitation performed well". There is no evidence found in the paper to prove that the rehabilitation performance here is good. Please provide an explanation.

6. It is suggested that the authors add comparisons with other robots (e.g., degrees of freedom, performance, etc.) to emphasize the innovation of this paper.

7. the introductory section is short and simply lists descriptions that do not serve as your summary of the shortcomings of the existing literature (lines 49-58)

8. The simulation part of this paper (Part VI) gives some simulation results, but also only gives the results and does a simple analysis,  further discussion is necessary for the simulation results.

9. The authors' work seems to have carried out the design, kinematic analysis and simulation of the ankle rehabilitation robot without conducting the corresponding experiments (even if tested on healthy subjects), and it is difficult to prove the conclusions of the authors of this paper by the simulation results alone.

10. The conclusions in the paper can prove the feasibility of the authors' work and do not seem to present what is the innovation of their work compared to other studies?

11. The references in the paper do not seem to provide a comprehensive summary of the relevant area. 

12. The English  in the paper seems to have some difficulty in understanding (e.g. lines 70-71, etc.), and it is recommended that the language should be checked.

13. The formulas are difficult to read, and the meaning of some unknown symbols is missing. For example, equations 1 and 2.

14. The content in Figure 6 is difficult to see and understand.

15. It is recommended that the condition numbers in Figure 12 be normalized. The numbers alone are actually difficult to understand the exact meaning.

Author Response

Reviewer 3

  1. This paper proposes a foot and ankle rehabilitation robot, presents some work, but the innovations in the paper are not clear. It is suggested to present it in the introduction with subheadings.

Answer:

Authors have proposed the innovation of the research work of this manuscript in introduction according the existing ankle rehabilitation robot limitations.

  1. Please explain the title of 3.1 (and 3.2)? Is it an inverse kinematic analysis?

Answer:

Author expressed the Jacobia matrix incorrectly in the inverse kinematic analysis, which issue has modified through the closed-loop vector method.

  1. It is suggested that the black curve in Figure 18 in the paper be in another color or line type, which is easily misunderstood. (Other figures as well)

Answer:

Authors have revised all of figures in rehabilitation exercise Section from Figure 18 to Figure 22.

  1. The description in line 404 of the text guarantees safety. It seems that the current results or conclusions in the paper is hard to support this conclusion.

Answer:

Authors have revised the conclusions based on the movements characterized by the study result.

The revised part was labelled in line from 459 to 463.

  1. In line 410, the author says "The results show that the ankle muscles were effectively exercised during rehabilitation and the ankle rehabilitation performed well". There is no evidence found in the paper to prove that the rehabilitation performance here is good. Please provide an explanation.

Answer:

Authors have revised the conclusions based on the movements characterized by the study result.

The revised part was labelled in line from 464 to 468.

  1. It is suggested that the authors add comparisons with other robots (e.g., degrees of freedom, performance, etc.) to emphasize the innovation of this paper.

Answer:

Author summarizes and improves the main innovation work and results of this, and specifically it from the abstract and conclusions.

  1. The introductory section is short and simply lists descriptions that do not serve as your summary of the shortcomings of the existing literature (lines 49-58).

Answer:

Authors have analyzed and commented on the ankle mechanisms citied in the introduction.

  1. The simulation part of this paper (Part VI) gives some simulation results, but also only gives the results and does a simple analysis, further discussion is necessary for the simulation results.

Answer:

The authors discuss the simulation results.

  1. The authors' work seems to have carried out the design, kinematic analysis and simulation of the ankle rehabilitation robot without conducting the corresponding experiments (even if tested on healthy subjects), and it is difficult to prove the conclusions of the authors of this paper by the simulation results alone.

Answer:

The authors performed the simulation analysis through the human-computer interaction software anybody.

  1. The conclusions in the paper can prove the feasibility of the authors' work and do not seem to present what is the innovation of their work compared to other studies?

Answer:

“The proposed ankle rehabilitation mechanism without axis offset and provide all degree of freedom according the ankle movement mode. Meanwhile, the robot can meet the size requirements of different adult lower limbs.” Which innovation was highlighted in introduction.

  1. The references in the paper do not seem to provide a comprehensive summary of the relevant area.

Answer:

Authors have analyzed and commented on the ankle mechanisms citied in the introduction,

  1. The English in the paper seems to have some difficulty in understanding (e.g. lines 70-71, etc.), and it is recommended that the language should be checked.

Answer:

The author has modified the manuscript, including grammar, case, and punctuation etc.

  1. The formulas are difficult to read, and the meaning of some unknown symbols is missing. For example, equations 1 and 2.

Answer:

The author has elaborated the formula 1 formula 2 and annotated the meaning of letters and symbols in line107-115 in revised manuscript.

  1. The content in Figure 6 is difficult to see and understand.

Answer:

The author introduced the 3D prototype into the ADAMS environment for the experimental environment of motion performance simulation. It has been detailed in the text.

  1. It is recommended that the condition numbers in Figure 12 be normalized. The numbers alone are actually difficult to understand the exact meaning.

Answer:

Figure 12 is from Matlab programming, which has been described in detail by the authors in the text.

Reviewer 4 Report

The manuscript delas with a very interesting topic in rehabilitation robotics. However, in my opinion, there are many aspects which must be improved before the article could be published. In specific:

1. Authors must specify the relevance of the cited references. For example, references [10]-[12] (and other) are cited in a very generic manner. It is not clear the importance of these reference in the context of the article. Reference [15] is not about any ankle rehabilitation robot. I don't understant the reason for citation this reference.

2. Authors affirm that "most robots have lower degrees of freedom...". However, most of the robots presented in cited references have 3 degrees of freedom which correspond with three rotation in the ankle joint. This affirmation is not clear.

3. In the first paragraph of section 2.1 inversion/eversion and abduction/adduction movements are describen referenced to incorrect axis (Y and Z axis are inverted).

4. In the first paragraph authors affirm: "Ankle movement forms suggested that can be considered as three rotation and one translation degrees of freedom in the mechanism". This affirmation must be clarified.

5. Notation and meaning of equations (1) and (2) are very confusing and difficult to follow.

6. In section 2.3 is necessary a more detailed description of the structural configuration of the robot. How much degrees of freedom it has? How many actuators are used? Which kind of actuators and where are they located?

7. In line 126, the reference must be to Fig. 3 instead Fig. 4.

8. In equation (11) the derivative with rescpect to time of the term including the rotation matrix is not zero. Pleas consult Section 4.2.1.1 in "Parallel Robots Mechanics and Control" from Hamid D. Taghirad.

9. In equation (8), what is d3? In Fig. 3 only d1 is shown and I can deduct location of d2, but not for d3.

10. In lines 103-105, authors must clarify how the universal joint help to keep coincident the rehabilitation mechanism with the ankle center.

11. The kinematic analysis development is not clear. In equation (3), author use a rotation matrix with three rotation angles, but in equation (17) only appears two angle rotation.

12. In rows 156-158 authors mention that the inverse position analysis is carried out by using equation (17). However, equation (17) gives a relationship between velocities and not between positions. This issue must be clarified. Moreover, the numerical parameters used to obtain results presented in Table 2 must be justified.

13. The obtained kinematic model (positions and velocities) must be validated by comparing them with simulation of the mechanism in ADAMS.

14. A general revision about grammar is needed.

Author Response

Reviewer 4

  1. Authors must specify the relevance of the cited references. For example, references [10]-[12] (and other) are cited in a very generic manner. It is not clear the importance of these reference in the context of the article. Reference [15] is not about any ankle rehabilitation robot. I don't understand the reason for citation this reference.

Answer:

Authors have analyzed and commented every reference about the ankle mechanisms citied in the introduction. which were marked in introduction. Reference [15] is spherical parallel manipulator revered the kinematic characteristics of the spherical mechanism.

  1. Authors affirm that "most robots have lower degrees of freedom...". However, most of the robots presented in cited references have 3 degrees of freedom which correspond with three rotations in the ankle joint. This affirmation is not clear.

Answer:

The authors have described this view in detail in line 68 in revised manuscript.

  1. In the first paragraph of section 2.1 inversion/eversion and abduction/adduction movements are describe referenced to incorrect axis (Y and Z axis are inverted).

Answer:

The author has corrected the representation of the coordinates according to Figure 1. In addition, the axis direction of the various ankle movements is identified in the figure 1.

  1. In the first paragraph authors affirm: "Ankle movement forms suggested that can be considered as three rotation and one translation degrees of freedom in the mechanism". This affirmation must be clarified.

Answer:

This part has been supplemented in line 99-102 in revised manuscript.

  1. Notation and meaning of equations (1) and (2) are very confusing and difficult to follow.

Answer:

The author has elaborated the formula 1 formula 2 and annotated the meaning of letters and symbols in line 107-115.

  1. In section 2.3 is necessary a more detailed description of the structural configuration of the robot. How much degrees of freedom it has? How many actuators are used? Which kind of actuators and where are they located?

Answer:

This part has been supplemented in line 127-145 in revised manuscript.

  1. In line 126, the reference must be to Fig. 3 instead Fig. 4.

Answer:

It has been corrected.

  1. In equation (11) the derivative with respect to time of the term including the rotation matrix is not zero. Pleas consult Section 4.2.1.1 in "Parallel Robots Mechanics and Control" from Hamid D. Taghirad.

Answer:

Section 3 analyzes the kinematics including inverse forward position analysis. Therefore, derivative is not necessary.

  1. In equation (8), what is d3? In Fig. 3 only d1 is shown and I can deduct location of d2, but not for d3.

Answer:

In order to specify the meaning of each letter, the author clearly marked each letter in Figure 3.

  1. In lines 103-105, authors must clarify how the universal joint help to keep coincident the rehabilitation mechanism with the ankle center.

Answer:

The issue was explicated in line 139-145 in revised manuscript.

  1. The kinematic analysis development is not clear. In equation (3), author use a rotation matrix with three rotation angles, but in equation (17) only appears two angle rotation.

Answer:

Formula (3) expresses the rotation transformation of the three degrees of freedom in the space, which is the general formula. According to the reviewer's suggestion, in order to express the research object of this paper, the author revised formula (3) according to the characteristics of the organization.

  1. In rows 156-158 authors mention that the inverse position analysis is carried out by using equation (17). However, equation (17) gives a relationship between velocities and not between positions. This issue must be clarified. Moreover, the numerical parameters used to obtain results presented in Table 2 must be justified.

Answer:

Section 3 is the kinematics analysis including inverse forward position. Therefore, derivative is not necessary. Authors have modified Section 3.

  1. The obtained kinematic model (positions and velocities) must be validated by comparing them with simulation of the mechanism in ADAMS.

Answer:

The author calculated the motion characteristics by changing the initial value of the genetic algorithm. The calculation results are shown in Table 2-3; Basically consistent with the simulation results in Figure 7 and Figure 9. Meanwhile, the author supplements this conclusion to the manuscript in line 280-281.

  1. A general revision about grammar is needed.

Answer:

Author carefully checked the grammar throughout the manuscript.

Round 2

Reviewer 2 Report

The revision is solid. No more comments. 

Reviewer 3 Report

The manuscript now can be accepted.

Reviewer 4 Report

Authors have attended all the comments from the previous revision.